# Spatial and Temporal Evolution of Ecosystem Service Values and Topography-Driven Effects Based on Land Use Change: A Case Study of the Guangdong–Hong Kong–Macao Greater Bay Area

Hui Li [1,2,*], Yilin Huang [1], Yilu Zhou [3], Shuntao Wang [1], Wanqi Guo [1], Yan Liu [1], Junzhi Wang [1], Qing Xu [1], Xiaokang Zhou [1], Kexin Yi [1], Qingchun Hou [1], Lixia Liao [1] and Wei Lin [1,*]

1   College of Forestry and Landscape Architecture, South China Agricultural University, Guangzhou 510642, China; 1664039245@stu.scau.edu.cn (Y.H.); wangshuntao@stu.scau.edu.cn (S.W.); guowanqi@stu.scau.edu.cn (W.G.); lynnl@stu.scau.edu.cn (Y.L.); wangjunzhi@stu.scau.edu.cn (J.W.); xuqing2022@stu.scau.edu.cn (Q.X.); zhouxiaokang@stu.scau.edu.cn (X.Z.); yikexin@stu.scau.edu.cn (K.Y.); hqc@stu.scau.edu.cn (Q.H.); liaolisa@stu.scau.edu.cn (L.L.)
2   Guangdong Rural Construction Research Institute, Guangzhou 510642, China
3   College of Architecture and Planning, Yunnan University, Kunming 650300, China; 18876638633@163.com
*   Correspondence: ydlihui@scau.edu.cn (H.L.); wlin@scau.edu.cn (W.L.)

**Abstract:** The Guangdong–Hong Kong–Macao Greater Bay Area (GBA) is rich in natural and marine resources, and it is scientifically valuable to study the evolution patterns and driving mechanisms of the ecosystem service values (ESVs) of the GBA for the governance and conservation of its ecosystems. Based on the land use changes in the GBA from 2000 to 2020, the ESVs in the GBA were measured at the grid scale, and the Markov model was used to predict the ESVs in 2030; the calculated results were used to analyze the spatial and temporal variation characteristics of the ESVs during the 30-year period, while the driving role of the topographic factors on the ESVs is revealed through the construction of the geographically weighted regression model (GWR). The results show the following: (1) During the 20-year period, the area of arable land and water in the GBA fluctuated greatly, with the area decreasing year by year and shifting mainly into construction land; in terms of shifting the center of gravity of the land, and the center of gravity of the grassland and unused land shifted the greatest distance due to the expansion of construction land, with the center of gravity shifting westward as a whole. (2) The ecosystem services (ESs) in the GBA show obvious aggregation in the spatial distribution, with the total ESVs decreasing year by year. Among them, the areas with an increasing total value are mainly located in the cities of Zhaoqing and Huizhou in the GBA, accounting for 27%, and the areas with a decreasing total value year by year are mainly located in the dense urban areas in the central part of the GBA, accounting for 35%, and the area is increasing, indicating that the habitat quality is deteriorating, and the model prediction shows that the value of ecosystem services in 2030 have a decreasing trend under the development of the natural state. (3) Topographic factors have a significant influence on the ESVs, and in terms of spatial distribution, the areas with the strongest effect are distributed in the northwestern and northeastern parts of the GBA, and the main uses for the land are wood land, arable land, water and the area of the water–land intersection near the sea.

**Keywords:** The Guangdong–Hong Kong–Macao Greater Bay Area; ecosystem service value; temporal and spatial evolution; topographic factors; geographically weighted regression model

## 1. Introduction

As a strategic national development region, the Guangdong–Hong Kong–Macao Greater Bay Area (GBA) has a key geographical location and a wealth of natural resources. The sustainable development of the GBA plays a crucial role in the enhancement of regional ESVs as well as the construction and optimization of ecological patterns [1]. For the coastal areas with complex and fragile environments, impaired habitat quality will lead

to a decline in the ESVs and negatively affect environmental security and socioeconomic development [2]. Therefore, strengthening the accounting of the ESVs and analyzing the influencing factors that affect the evolution of the ESVs will help with the governance and protection of ecosystems in the GBA [3].

The concept of ecosystems was first proposed by Tasley in the 1930s, marking the gradual refinement of the study of ecosystems; on this basis, SCEP first proposed the concept of ESs in the 1970s [4], and listed a number of ESs provided by the natural environment. Subsequently, Costanza deepened his research on ESVs and classified them into 17 categories, including gas regulation, climate regulation, water regulation, soil formation, nutrient cycling and recreation, based on four dimensions including production, basic functions, environmental benefits and recreation, which became the basis for the classification of subsequent scholars and greatly enriched the research results in this field. At the beginning of the 21st century, the UN Millennium Ecosystem Assessment redefined the concept of ESs and classified them as provisioning, regulating, cultural and supporting services in relation to human benefits, leading to a boom in research on the value of ecosystem services. In China, the study of ESs began in the 1990s when Liu Xiaodi introduced the concept of ESs to the country based on the research results of Daily [5]; subsequently, Ouyang Zhiyun conducted in-depth research on this concept, continuously enriching the theoretical findings on ecosystem services [6]. In the same era, Xie Gaoqi defined ES functions as products and services obtained directly or indirectly through ecosystem functions and classified them into three major categories, namely, production functions, ecological functions and recreational and leisure functions [7]; since then, domestic research on ES functions has been gradually enriched and diversified [8,9].

Early research on ESs at home and abroad focused on the valuation of ecosystem services. There are two main approaches to accounting; the first is the physical quality assessment method, which, in turn, includes the functional volume assessment method and the energy value assessment method. The material quality refers to the value of the final products or services obtained by humans directly or indirectly from the ecosystem [10]. Its principle is to obtain the total value of an ecosystem service from its functional volume and the unit price of that volume [11,12]; the functional volume of the relevant indicator is usually calculated using models such as InVEST and ARIES, which are commonly used. The advantages of the physical quality assessment method are that the results can objectively reflect the structural functions and ecological processes of the ecosystem, but the limitations are that there are uncertainties in the data acquisition and the calculation process is cumbersome, and there are differences in the units of measurement for the individual functional indicators, which makes it difficult to measure the value of ecosystem services with multiple functions. The second approach is the value–volume approach, which includes the monetary and value-equivalent approaches. The principle of this approach is to quantify ecosystem services using economic algorithms, including the direct market approach, the substitution market approach and the simulated market approach [13]. The analysis of the different ecosystem service valuation methods shows that each method has its own advantages and limitations, so the appropriate method can be weighed according to the purpose and focus of the study [1]. Based on changes in the social environment and the different methods of accounting for value, scholars measured the ESVs at different environmental scales and gradually built up a systematic research method for quantifying ESs [14–17], of which a highly representative research result was the principle and method of valuing ESs proposed by Costanza et al. in 1997 [18]; Costanza first accounted the ESVs at the global scale using this method. Subsequently, Chinese scholars such as Xie Gaodi combined the actual situation of ESs in China and, after a series of revisions, developed a table of value equivalent factors applicable to terrestrial ecosystems in China [19]. Later research directions integrate spatial techniques in value assessment to explore the coupling between the land use types and the ESVs. The number and rate of land type changes can directly reflect the intensity of the land type conversion [20–22], thus exploring the temporal and spatial evolution and variation of ESVs in the region from multiple dimensions.

At present, research on the factors influencing the evolution of ESVs has made great progress, mainly focusing on the relevance of physical geographic factors to ESVs [23–26], while regional topography, as one of the determinants of landscape patterns, has an important influence on the evolution of ESVs [27,28]. There are two main methods for coupling ESVs with topographic factors. The first is the direct coupling method, which is mainly calculated through the topographic position index; this index is a composite analysis of the elevation and slope, and is often used to quantify the spatial effects of land use on topographic gradients. The second is the model coupling method, which is mainly carried out through a spatial autocorrelation analysis, such as the Moran index and the geographically weighted models, among which the GWR is more advantageous in the analysis of coupling relationships, because it can express the data characteristics of spatial data at different locations and can reflect the weight size of local areas in the data [29,30]. Therefore, it is important to use spatial autocorrelation analysis to study the driving effect of ESVs in order to optimize the regional ESVs and build an ecological security pattern [31,32].

From the current research results, it can be seen that studies on ESVs in the GBA mostly focus on the relationship between urbanization and ESVs, but the depth of the studies are still insufficient [33–35], mainly in two aspects. Firstly, current studies on the spatial and temporal variation of regional ESVs seldom involve future-year scenarios, and at the same time, when analyzing the spatial and temporal variation of ESVs in regions with rapid urbanization, the coupling study of ESVs and land use changes is seldom involved; secondly, studies of the driving effects of ESVs within the GBA are less likely to involve topographic factors. Therefore, based on the remote sensing image interpretation maps of the GBA from 2000 to 2020, this study measures and predicts ESVs in the GBA in 2030 at the grid scale, analyzes its spatial and temporal variation characteristics, and introduces a model called GWR to reveal the driving effect of topographic factors on ESVs. The aim is to explore the following issues: (1) the spatial and temporal evolutionary characteristics of ESVs in the GBA; (2) the impact of land use on changes in the ESVs; and (3) the relationship between topographic factors and ESVs. These issues are explored to deepen the content of the study and to provide theoretical guidance for the GBA ecosystem to achieve regional ecological governance and integrated urban development.

## 2. Materials and Methods

### 2.1. Study Area Overview

The GBA belongs to one of the four major bay areas in the world, with a total surface area of about 56,000 square kilometers. In terms of administrative area, it mainly consists of the nine cities of Guangzhou, Shenzhen, Zhuhai, Foshan, Huizhou, Dongguan, Zhongshan, Jiangmen and Zhaoqing in the Guangdong Province and the special administrative regions of Hong Kong and Macau (Figure 1). The study area has a mild and humid climate, with a dense network of rivers and water, and is distributed in the mountainous forests of Mafeng Mountain–Baiyun Mountain, Gudou Mountain–Wuguishan Mountain–Phoenix Mountain, Daling Mountain–Yangtai Mountain–Tanglang Mountain, etc., which form an inter-urban ecological transition zone. The study area also has a wide distribution of forests and agricultural land, with arable land scattered throughout the bay area; wetland resources are abundant, with water entering the rivers located in the four cities of Foshan, Zhuhai, Shenzhen and Zhongshan [36,37].

### 2.2. Data Sources and Initial Data Processing

The remote sensing image source data selected for this study were obtained from the Resource and Environment Science and Data Centre of the Chinese Academy of Sciences (http://www.resdc.cn/, accessed on 1 July 2021), with three periods including 2000, 2010 and 2020, and the spatial resolution of the data is 30 m. The topographic data were obtained from the geospatial data cloud database (http://www.gscloud.cn/, accessed on 1 July 2021), with the type ASTER GDEM. The relevant statistical data were mainly obtained from the Guangdong Statistical Yearbook, the Hong Kong Statistical Yearbook and the

Macao Statistical Yearbook during the study period. The collected remote sensing data were corrected and classified using ArcGIS software, and according to the previous research results, the land types were classified into six categories, such as construction land, arable land, grassland, woodland, water and unused land [38]; the relevant yearbook data were collated, summarized and analyzed, and processed using SPSS software.

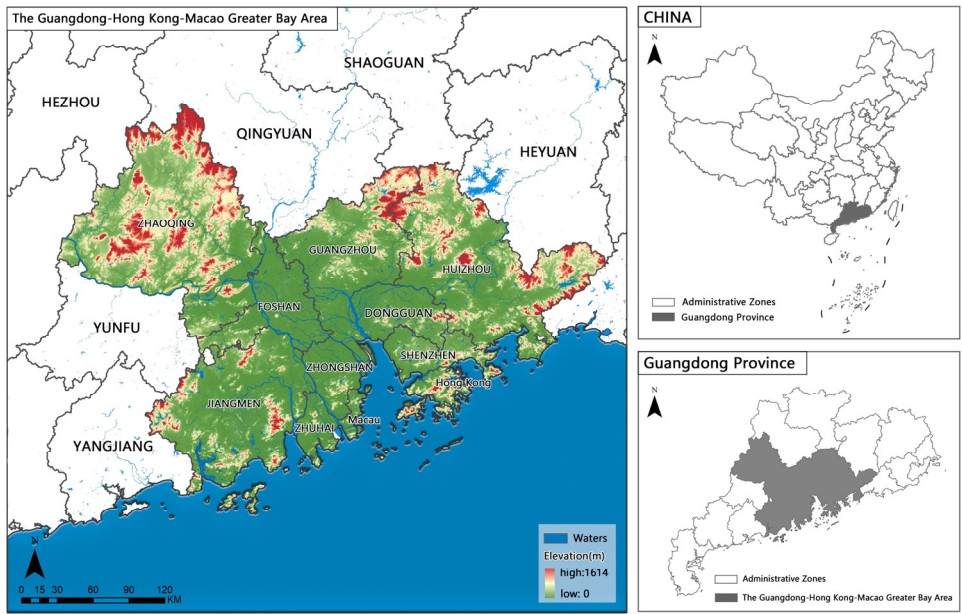

**Figure 1.** Map of the Guangdong–Hong Kong–Macao Greater Bay Area, China.

*2.3. Research Methods*

The research idea and process are shown in Figure 2.

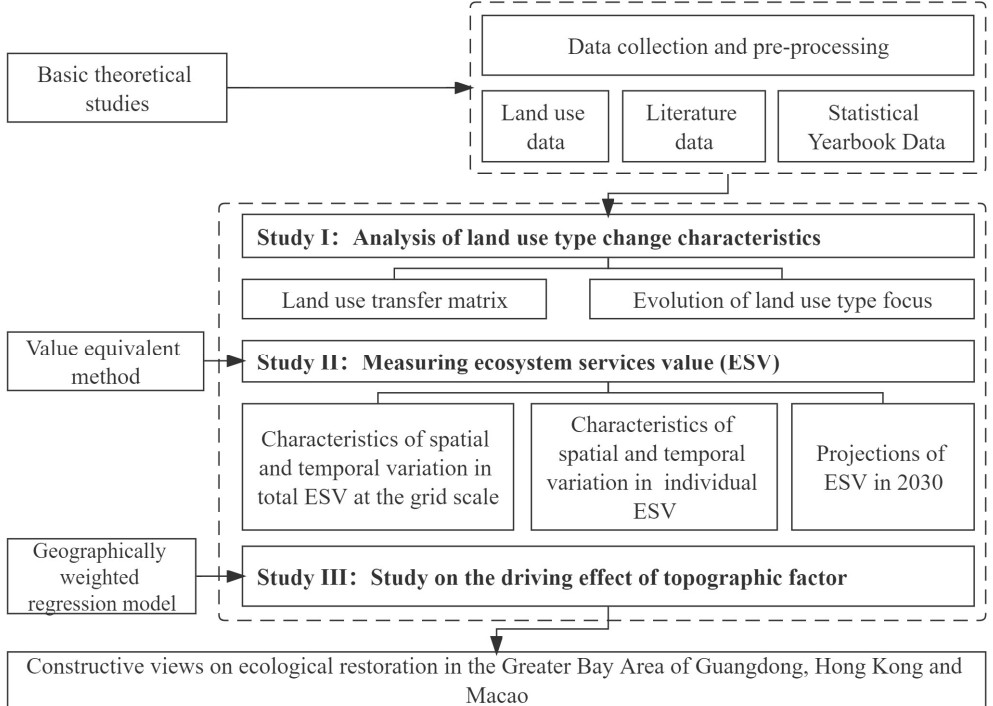

**Figure 2.** Article flow chart.

### 2.3.1. Analysis of Land Use Change

The land use transfer matrix can concretely reflect the characteristics of the land use structure. It reflects the transition relationship between a certain land type within a certain time interval, and calculates the transition process of land state from moment *t* to moment *t* + 1 using Equation (1) [39,40]; the direction and distance of the land center of gravity shift can reflect the natural economic conditions as well as the quality of land in the region [41,42], and the coordinate change of the land use type center of gravity can be calculated using Equation (2) as follows:

$$S_{ab} = \begin{bmatrix} S_{11} & S_{12} & \cdots & S_{1n} \\ S_{21} & S_{22} & \cdots & S_{2n} \\ \vdots & \vdots & \vdots & \vdots \\ S_{n1} & S_{n2} & \cdots & S_{nn} \end{bmatrix} \tag{1}$$

In the equation, $S_{ab}$ represents the land use status at the beginning and end of the study, and *n* represents the number of land use types.

$$X_a = \frac{\sum\limits_{i=1}^{n}(C_{ai} \times X_i)}{\sum\limits_{i=1}^{n}C_{ai}} Y_a = \frac{\sum\limits_{i=1}^{n}(C_{ai} \times Y_i)}{\sum\limits_{i=1}^{n}C_{ai}} \tag{2}$$

In the equation, $X_a$ and $Y_a$ represent the latitude and longitude coordinates of the center of gravity of a land type in year *a*, $C_{ai}$ represents the area of the *i* patch in a land type in year *a*, and $X_i$ and $Y_i$ represent the latitude and longitude coordinates of the center of gravity of the *i* patch in a land type.

### 2.3.2. Ecosystem Service Value at the Grid Scale

The grid scale can quantify the spatial heterogeneity of the land use types in the study area and express their information adequately, which is an effective evaluation unit for quantifying the spatial and temporal evolution of the land use [43]. In this study, 1 km × 1 km, 3 km × 3 km, 5 km × 5 km, 10 km × 10 km, 15 km × 15 km and 20 km × 20 km grids were constructed as pre-selected evaluation units based on previous research results. Considering the scale and shape characteristics of the study area, in order to make the grid cells not only have certain regional characteristics, but also contain sufficient research information, and at the same time, be able to reflect the spatial variation of ESVs in the study area more freshly, we referred to a large amount of the literature relevant to our study and compared the spatial variation effect of different grid calculation results, and finally chose 3 km × 3 km as the base measurement cell (Figure 4). The calculation of ESVs referred to the equivalence factor table modified by Xie Gaodi according to the actual situation in China; in addition, considering the differences in grain prices in different regions, the average grain price of farmland in the Guangdong Province was chosen as the average grain price in this study, while the ESV coefficients for construction land referred to the coefficients proposed in the table for the value of each ecosystem service per unit area of the different types of terrestrial ecosystems in China to make corrections [44,45]. In terms of the value per unit area, ESVs at the grid scale were corrected using the coefficient proposed by Xie Gaodi (1.40 for Guangdong Province) and using the vegetation cover [46], resulting in ESVs per unit area in the GBA (Table 1), with the following revised equation:

$$FV = \frac{NDVI - NDVI_{\min}}{NDVI_{\max} - NDVI_{\min}}$$
$$F_a = f_{ai}/f \tag{3}$$
$$E = E_a \times F_a$$

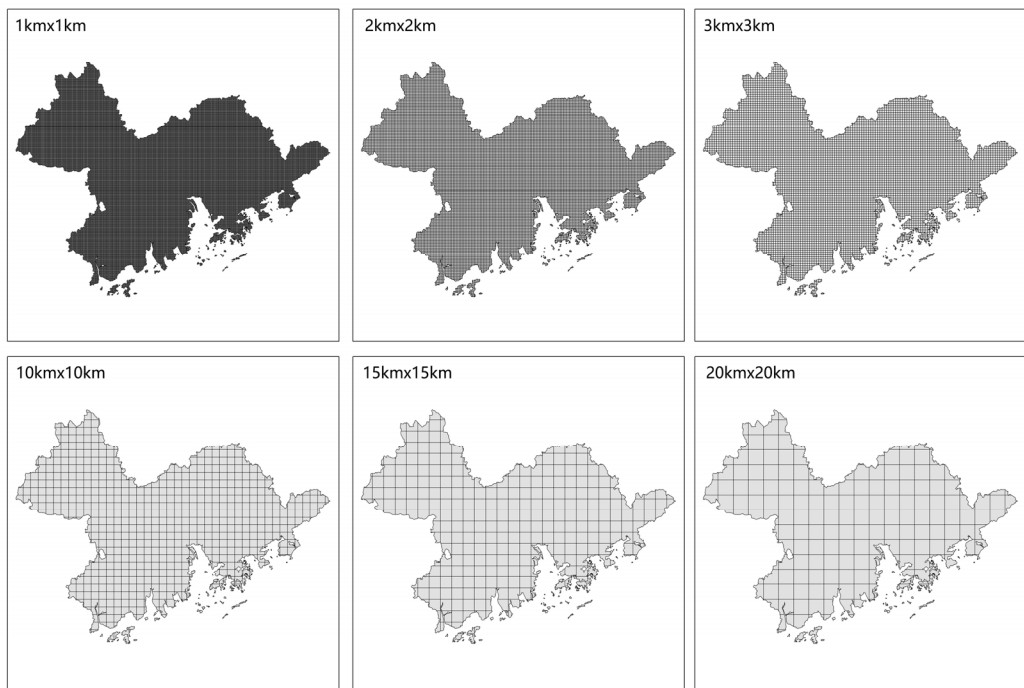

**Figure 3.** Schematic diagram of grid scale in the research area.

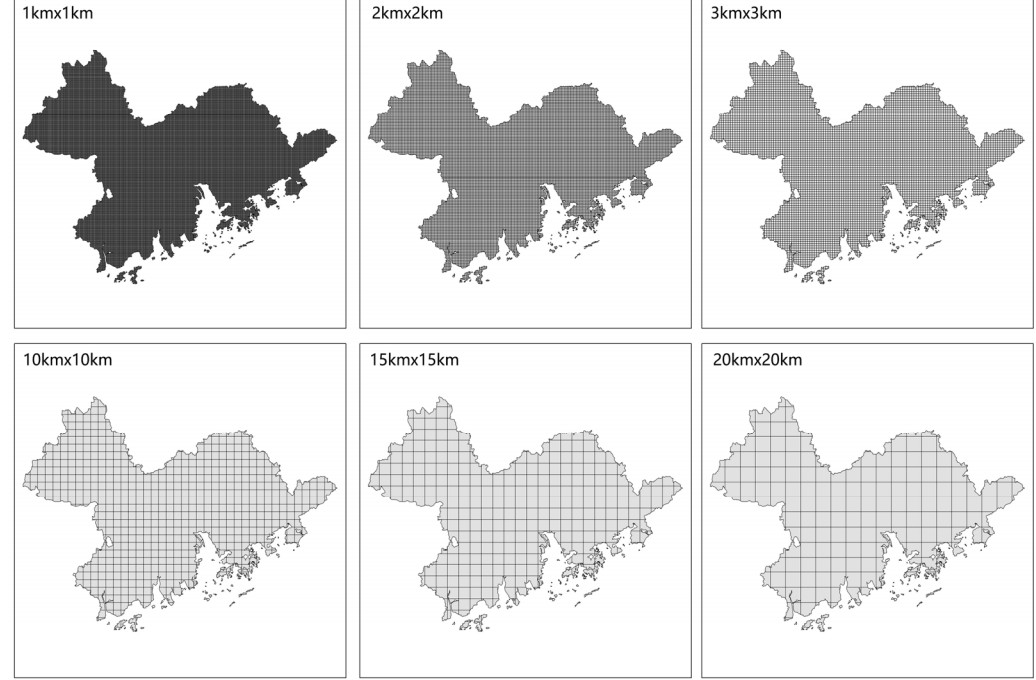

**Figure 4.** Schematic diagram of grid scale in the research area.

**Table 1.** Ecosystem service values ($1 \times 10^8$ CNY) per unit area in the Guangdong–Hong Kong–Macao Greater Bay Area.

| Ecosystem Services | Arable Land | Woodland | Grassland | Water | Unused Land | Construction Land |
|---|---|---|---|---|---|---|
| Food production | 1833.63 | 605.10 | 788.46 | 971.82 | 36.67 | - |
| Raw material production | 715.12 | 5464.22 | 660.11 | 641.77 | 73.35 | - |
| Atmospheric regulation | 1320.21 | 7921.28 | 2750.44 | 935.15 | 110.02 | - |
| Climate regulation | 1778.62 | 7462.87 | 2860.46 | 3777.28 | 238.37 | - |
| Hydrological regulation | 1411.90 | 7499.55 | 2805.45 | 34,417.24 | 128.35 | −6678.00 |
| Waste disposal | 2548.75 | 3153.84 | 2420.39 | 27,229.41 | 476.74 | −2174.10 |
| Soil maintenance | 2695.44 | 7371.19 | 4107.33 | 751.79 | 311.72 | 3480.00 |
| Biodiversity maintenance | 1870.30 | 8269.67 | 3428.89 | 6289.35 | 733.45 | - |
| Recreation | 311.72 | 3813.95 | 1595.26 | 8141.32 | 440.07 | - |
| Total | 14,485.68 | 51,561.68 | 21,416.80 | 83,155.12 | 2548.75 | −5372.10 |

In the equation, *FV* is the vegetation index, *NDVI* is the standard value of vegetation cover, $f_a$ denotes the sum of the *FV* values of the ath grid cell, $f$ is the mean value of *FV*, $E_a$ and $E$ refer to ESVs before and after the revision and $F_a$ denotes the vegetation cover revision factor of the ath grid cell.

### 2.3.3. Markov Model

Markov chain represents the sequential state of the development process of things, which is discrete in time and state, and the development process of things is only linked to adjacent things. Markov model can be used on this basis to predict the future state of things in the same time interval according to the development process and state of things [47–49]. In this study, the Markov model is used to predict and analyze the change in land use types in the GBA in 2030 and estimate the ESVs in the study area based on the prediction results. The relevant calculation equation is as follows:

$$S(a+1) = P_{ij} \times S(a) \tag{4}$$

In the equation, $S(a)$, $S(a+1)$ represent the states of the ground class at moments $a$, $a+1$; $P_{ij}$ represents the state transfer matrix.

### 2.3.4. Geographically Weighted Regression Model

Geographically weighted regression model (GWR) is a local regression model that assists the original spatial data in determining the coordinate location parameters [50], which is based on the principle of independent linear regression calculations at all point locations, thereby allowing for the expression of data characteristics of spatial data at different locations, perfectly compensating for the shortcomings of the global regression model, while reflecting the magnitude of the weights of local areas in the data [51,52]. In this study, the relationship between topography and slope data and ESVs is analyzed comprehensively by constructing a GWR model, in which topography and slope data are used as independent variables, and the ESVs are used as dependent variables, from which the corresponding explanatory variable coefficients (predicted) and local $R^2$ data can be calculated. The Y value represents the predicted estimate of the dependent variable, and the closer the Y value is to the ESVs, the better the fit between the two; the local $R^2$ means the fit of the local regression model to the Y value, with larger values indicating a higher model accuracy [53]. The relevant calculation equation is as follows:

$$Y_i = \beta_0(\mu_a, v_a) + \sum_{a=1}^{k} \beta_k(\mu_a, v_a) X_{ak} + \varepsilon_a \tag{5}$$

In the equation, $(u_a, v_a)$ represents the spatial coordinate point *a*; *Ya* and $K_{ak}$ are the dependent variable *Y* and the set of sub variables $X_k$ of the measured values of the spatial

position; $k$ is the number of independent variables; $\beta_0 (u_a, v_a)$ is the spatial position of the constant term $(u_a, v_a)$; $\beta_k (u_a, v_a)$ is the value of the continuous function $\beta_k (u_a, v_a)$ at point $a$ and $\varepsilon_a$ represents the random error term.

## 3. Results

### 3.1. Analysis of Spatial and Temporal Land Use Change Characteristics

3.1.1. Land Use Transfer Matrix

From the land use transfer matrix of the study area (Table 2), the urban boundaries within the GBA expanded continuously from 2000 to 2010, encroaching on the surrounding arable land, woodland and water. The areas of arable land, water and woodland transferred out of the GBA during the 10-year period were 2636 km$^2$, 1098 km$^2$ and 992 km$^2$, respectively, while the largest amount of construction land was transferred in, accounting for 43% of the original area, which is mainly due to the fact that the urban master plan expanded the use of reserve land resources such as mudflats and coastal zones, while the construction of infrastructure also accelerated the change in the construction land. From 2010 to 2020, arable land, waters and woodland remained the largest land types transferred out, with areas of 3459 km$^2$, 2141 km$^2$ and 1937 km$^2$, respectively, while construction land was transferred in at 4205 km$^2$. In general, the area of arable land in the GBA fluctuated due to the national policy of returning farmland to forests and the continuous reduction in the primary industry over the past 20 years; the area of woodland changed less, while the area of land transferred for construction reached 51% of the original area over the past 20 years, which shows that the rapid development of the GBA has put the ecological land under pressure (Figure 5).

**Table 2.** Land Use Transfer Matrix for the Greater Bay Area of Guangdong, Hong Kong and Macau (km$^2$).

| 2000 | 2010 | | | | | |
|---|---|---|---|---|---|---|
| | **Arable Land** | **Woodland** | **Grassland** | **Water** | **Construction Land** | **Unused Land** |
| Arable land | 11,794.79 | 200.47 | 6.57 | 640.16 | 1788.17 | 0.14 |
| Woodland | 146.54 | 29,617.05 | 23.74 | 65.25 | 755.93 | 0.39 |
| Grassland | 10.14 | 76.23 | 1063.23 | 11.52 | 64.40 | 0.03 |
| Water | 542.56 | 39.28 | 4.26 | 3152.38 | 511.80 | 0.03 |
| Construction land | 130.03 | 92.71 | 3.79 | 71.54 | 4135.00 | 0.05 |
| Unused land | 2.10 | 0.91 | 0.02 | 3.23 | 6.61 | 10.63 |
| 2010 | 2020 | | | | | |
| | **Arable Land** | **Woodland** | **Grassland** | **Water** | **Construction Land** | **Unused Land** |
| Arable land | 10,971.15 | 760.73 | 36.61 | 244.38 | 2407.70 | 9.73 |
| Woodland | 617.77 | 28,671.65 | 221.40 | 115.43 | 973.95 | 8.69 |
| Grassland | 42.13 | 270.49 | 820.73 | 11.59 | 79.91 | 0.71 |
| Water | 1169.43 | 144.25 | 11.69 | 2108.89 | 736.71 | 79.34 |
| Construction land | 232.96 | 122.17 | 5.63 | 34.43 | 4026.30 | 11.63 |
| Unused land | 3.44 | 1.85 | 0.14 | 0.18 | 6.74 | 11.17 |

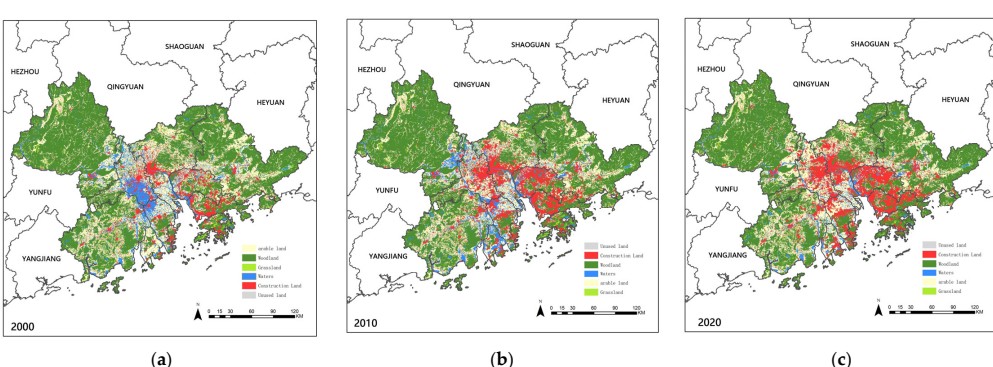

(a) (b) (c)

**Figure 5.** Land use interpretation map of Guangdong, Hong Kong and Macao Greater Bay Area in 2000 (**a**), 2010 (**b**) and 2020 (**c**).

3.1.2. Shift in Land Use Focus

From 2000 to 2020, the center of gravity of the grassland, unused land and water areas in the GBA shifted significantly. The main reason for this is the expansion of the urban land area and the excessive artificial reclamation, which led to the occupation of ecological land such as ponds, dike-ponds and lakes; the center of gravity of the grassland moved 22,897 m to the northwest, and the center of gravity of the urban construction land moved 3602 m to the north and then 936 m to the southeast. The results of the analysis show the impact of the reorientation of urban development in different periods on various types of land, and that the shift in the center of gravity of the land use indicates a change in the structure of the land use in the region, as the environmental carrying capacity and self-regulating ability of the GBA are affected, resulting in a change in the ESVs (Table 3).

**Table 3.** Land use focus shift in the Guangdong, Hong Kong and Macao Greater Bay Area (m).

| Type of Land Use | 2000–2010 | | 2010–2020 | |
|---|---|---|---|---|
| | **Transfer Distance** | **Transfer Direction** | **Transfer Distance** | **Transfer Direction** |
| Arable land | 3733 | Northwest | 3625 | West |
| Woodland | 1171 | Northwest | 591 | Southeast |
| Grassland | 2935 | Northwest | 19,962 | Northwest |
| Water | 3448 | South | 4276 | Southeast |
| Construction land | 3602 | North | 936 | Southeast |
| Unused land | 11,086 | East | 24,404 | Southwest |

*3.2. Analysis of the Spatial and Temporal Evolution of Ecosystem Service Values*

3.2.1. Time Series Changes in Ecosystem Service Values

In terms of the changes in the total ESVs, there is an overall decreasing trend from 2000 to 2020; in terms of the magnitude of change, there is a greater change in the waters, woodlands and croplands, thus indicating a continuous decline in the environmental quality of the region (Table 4).

**Table 4.** Changes of ecosystem service values ($1 \times 10^8$ CNY) in the Guangdong–Hong Kong–Macao Greater Bay Area.

| Ecosystem Service Functions | | 2000 | | | 2010 | | | 2020 | | |
|---|---|---|---|---|---|---|---|---|---|---|
| | | ESV | Contribution Rate (%) | Class | ESV | Contribution Rate (%) | Class | ESV | Contribution Rate (%) | Class |
| Supply Services | Food production | 50.55 | 2.33 | 9 | 46.69 | 2.25 | 9 | 45.29 | 2.36 | 9 |
| | Raw material production | 182.23 | 8.41 | 7 | 177.95 | 8.56 | 7 | 175.06 | 9.12 | 7 |
| Reconciliation Services | Atmospheric regulation | 270.50 | 12.48 | 5 | 263.59 | 12.68 | 5 | 259.44 | 13.52 | 4 |
| | Climate regulation | 275.75 | 12.72 | 4 | 267.60 | 12.87 | 4 | 258.97 | 13.50 | 5 |
| | Hydrological regulation | 377.05 | 17.40 | 1 | 344.90 | 16.59 | 1 | 277.16 | 14.44 | 3 |
| Support Services | Waste disposal | 247.68 | 11.43 | 6 | 229.28 | 11.03 | 6 | 180.65 | 9.41 | 6 |
| | Soil maintenance | 289.92 | 13.38 | 3 | 289.89 | 13.94 | 3 | 290.58 | 15.14 | 2 |
| | Biodiversity maintenance | 313.71 | 14.48 | 2 | 304.34 | 14.64 | 2 | 291.28 | 15.18 | 1 |
| Cultural Services | Recreation | 159.83 | 7.37 | 8 | 155.27 | 7.47 | 8 | 140.34 | 7.31 | 8 |
| | Total | 2167.22 | 100 | - | 2079.50 | 100 | - | 1918.76 | 100 | - |

In terms of the change in the ESVs of the different land types, hydrological regulation, biodiversity and soil conservation have the highest contribution ratings, with all three contributing more than 44.8%, while the least contribution is made by recreation and food production, which account for less than 10%. Specifically, soil conservation, food production and biodiversity maintenance are the main service functions of arable land; the main service function of woodland is biodiversity; the main service function of grassland is soil conservation, but the overall contribution of grassland is low due to its small footprint and the main service function of the watershed area is hydrological regulation and waste disposal. It is worth mentioning that the ESVs of the watershed area declined significantly by $139.37 \times 10^8$ CNY from 2010 to 2020, which is mainly due to the destruction of watershed habitats as a result of the dike-pond reclamation project and urban expansion, with the watershed area decreasing year by year. Overall, the ESVs in the study area show a decreasing trend due to the expansion of urban construction sites and the increasing negative impacts of human disturbances such as industrial waste emissions, domestic waste and vehicle exhaust (Figure 6).

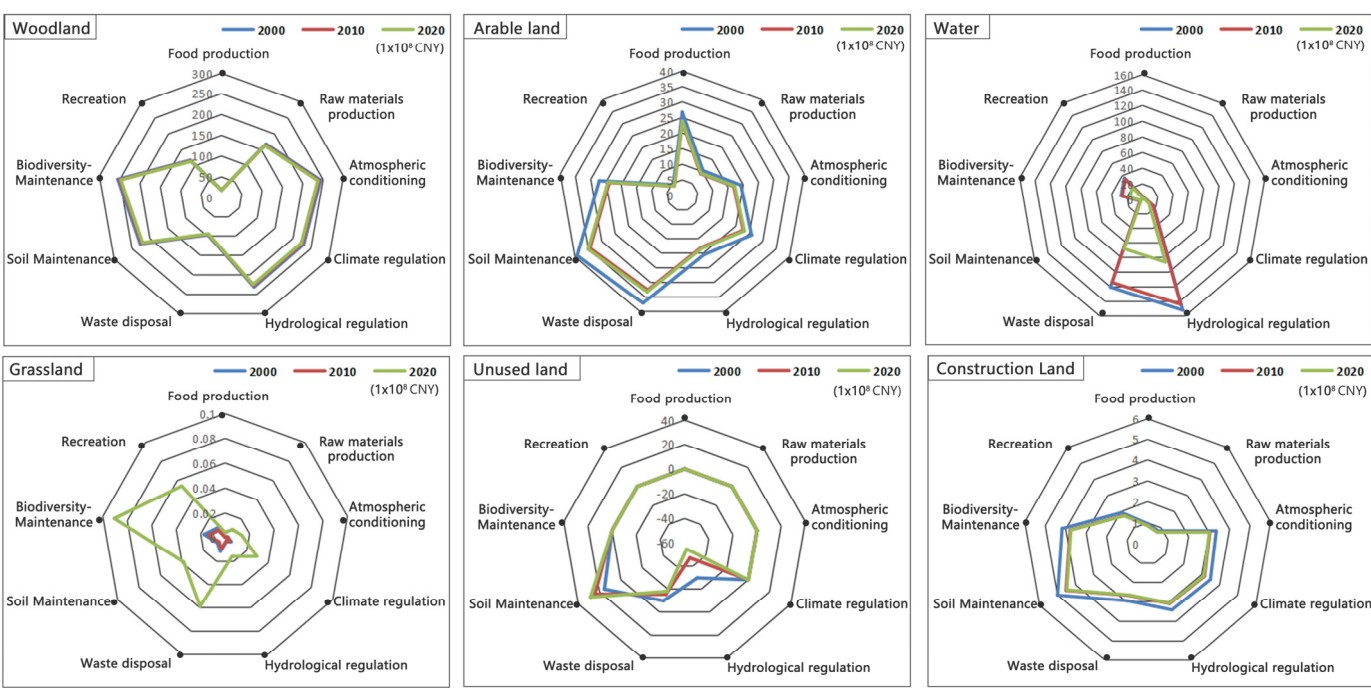

**Figure 6.** Individual ecosystem service values in the Guangdong–Hong Kong–Macao Greater Bay Area.

### 3.2.2. Spatial Changes in Ecosystem Service Values

The results of ESVs in the GBA are classified into the following five standard classes: very low(ESV < $15 \times 10^6$ CNY/hm$^2$), low (ESV of $15 \times 10^6$–$30 \times 10^6$ CNY/hm$^2$), medium (ESV of $30 \times 10^6$–$45 \times 10^6$ CNY/hm$^2$), high (ESV of $40 \times 10^6$–$60 \times 10^6$ CNY/hm$^2$) and very high (ESV > $60 \times 10^6$ CNY/hm$^2$). In terms of the spatial distribution of the regional ESV class (Figure 7), the areas with a very high ESV class in the GBA are mainly located in the northeastern part of Zhaoqing, the northeastern part of Huizhou and the western and southern part of Jiangmen, which is mainly due to the rich woodland resources and good ecological substrate in the area. The very low ESV classes are found in the central part of the region, which is mainly due to the concentration of urban agglomerations in the region and the high level of disturbance from human activities, which accelerated changes in vegetation, climate and other related factors and led to a continuous decrease in the quality of water areas and water resources, with a large number of ponds and other water areas being encroached upon, as well as the negative impact of urban sewage on rivers and marine ecosystems, ultimately leading to a continuous decrease in the quality of habitats in the central urban agglomerations. Specifically, from 2000 to 2010, the area of low ESVs in the GBA expanded significantly. By 2020, low

ESV areas became the dominant class type, accounting for 27.24% of the area, and still show an increasing trend, indicating that the ESVs in the GBA have been decreasing over the past 20 years. From the grid cells, the ESV cells in the GBA are mainly medium and low grade, and the high ESV area shows a small increase between 2000 and 2020, from 1534 cells to 1740 cells, which is mainly due to the related cities in the northwest of the GBA to increase the protection and restoration of forest land, slowing down the trend of reducing the forest land area. Among them, the proportion of high-grade cells in the Hong Kong and Macao special administrative regions is 3%, and it shows a small increase over 20 years, which is mainly due to the fact that the forest area in the Hong Kong and Macao special administrative region has increased by about 85 km$^2$ in 20 years, which contributed to the rebound of ESVs to a certain extent (Table 5).

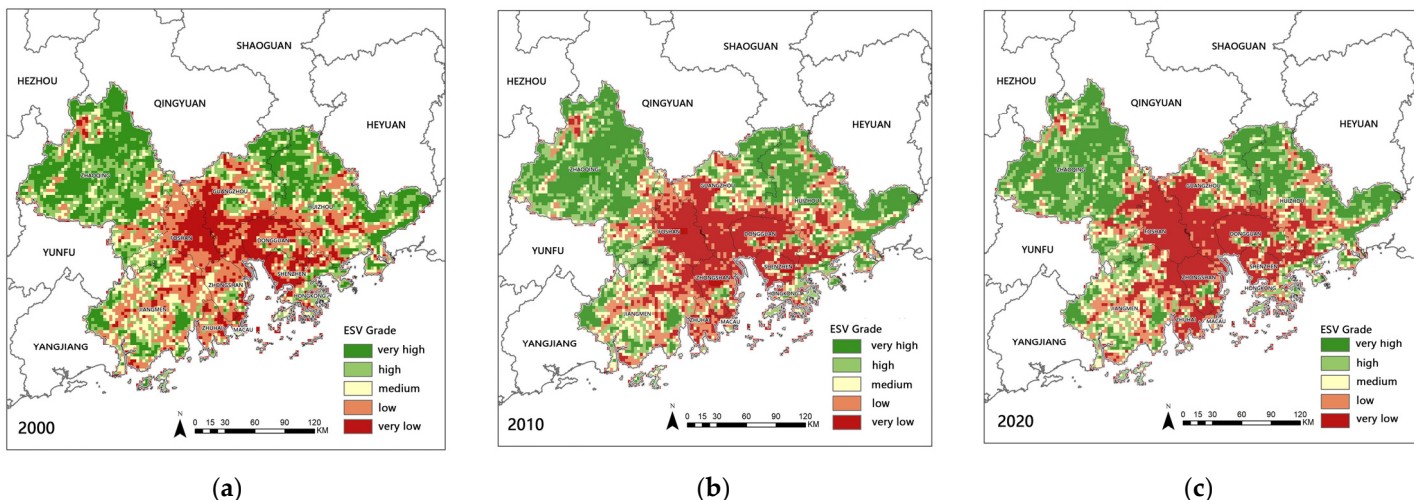

**Figure 7.** Ecosystem service values class for the Guangdong–Hong Kong–Macao Greater Bay Area in 2000 (**a**), 2010 (**b**) and 2020 (**c**).

**Table 5.** Area statistics for different classes of ecosystem service values (number of cells).

| Year | Very Low | Low | Medium | High | Very High |
|------|----------|------|--------|------|-----------|
| 2000 | 1092 | 1577 | 1091 | 1157 | 1534 |
| 2010 | 1543 | 1159 | 933 | 1055 | 1761 |
| 2020 | 1757 | 992 | 916 | 1046 | 1740 |

In terms of the spatial changes in the ESV class, the overall ESV rank in the GBA remained stable from 2000 to 2010, with 34% of the regional ranks increasing, and 29% of them decreasing, mainly because the economic growth in the GBA was still slow at the beginning of the 21st century, with less disturbance to the ecological environment and land. Among them, the nine cities in the Pearl River Delta have a high proportion of areas with no significant changes and small increases in the ES levels, while the Hong Kong and Macao special administrative regions have an increasing trend in the ES function levels, mainly because the ecological restoration of Mai Po Nature Reserve in Hong Kong was effective, and the restored mangrove and mudflat ecosystems can provide a better ecological benefit. During 2010–2020, the overall ES level of the GBA declined significantly, among which the ES level of nine cities in the Pearl River Delta declined significantly, which is mainly due to the continuous adjustment in urban construction and industrial structure, and the accelerated growth rate of economic development in Zhaoqing, Huizhou, Zhongshan and other cities. A total of 33% of the regional ESV levels in the Hong Kong and Macau special administrative regions showed a declining trend, among which the level in the coastal areas declined significantly, which is mainly due to the prosperity of land reclamation projects in coastal areas, the decline of the retention rate of the natural shoreline, and the large changes

in the regional ecological environment quality and ecological pattern (Table 6). Overall, 38% of the regional ESV class in the GBA did not change significantly over the 20-year period, the proportion of decreasing regional cells was 35%, the proportion of increasing regional cells was 27% and the number of decreasing regional cells continuously increased, indicating that the regional habitat quality is deteriorating (Figure 8).

**Table 6.** Area statistics on changes in ecosystem service value class (number of cells).

| Year | ESV Reduction Zone ($<-2.4 \times 10^6$ CNY/hm$^2$) | ESV No Significant Change Zone ($-2.4$–$2.4 \times 10^6$ CNY/hm$^2$) | ESV Increase Zone ($>2.4 \times 10^6$ CNY/hm$^2$) |
|---|---|---|---|
| 2000–2010 | 1875 | 2387 | 2189 |
| Proportion | 29 | 37 | 34 |
| 2010–2020 | 2098 | 3373 | 980 |
| Proportion | 33 | 52 | 15 |
| 2000–2020 | 2286 | 2436 | 1729 |
| Proportion | 35 | 38 | 27 |

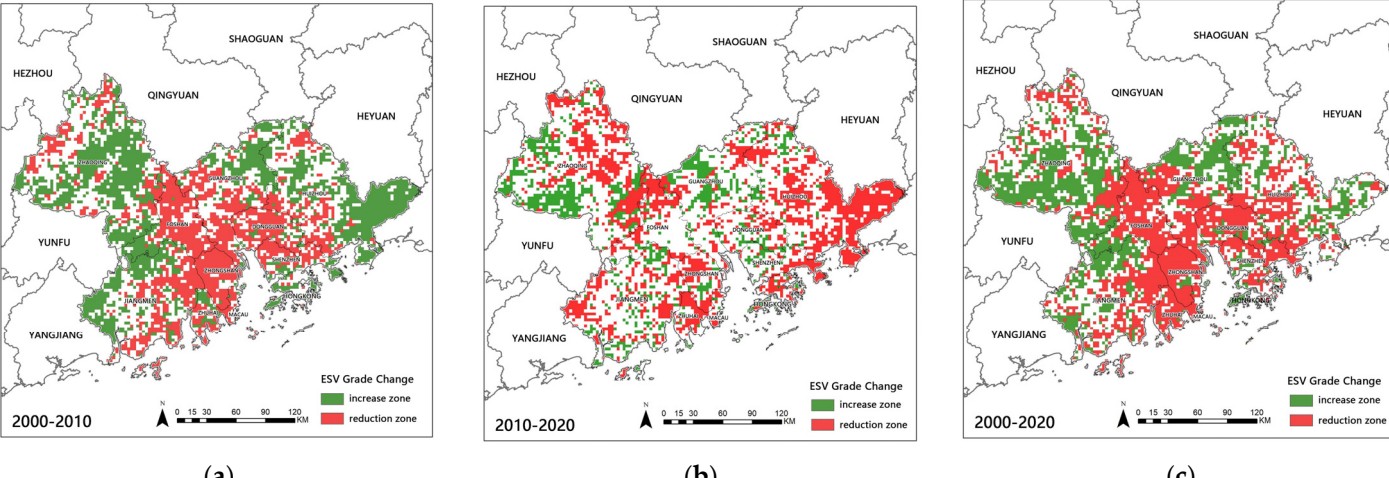

**Figure 8.** Areas of change in ecosystem service values class in 2000 (**a**), 2010 (**b**) and 2020 (**c**).

In terms of the regional spatial correlation, the ESVs in the GBA show significant spatial aggregation, with "high-high" and "low-low" indicating the proximity of high-value and low-value areas of ESVs. The global Moran index of the study area shows an increasing trend from 2000 to 2020, indicating the increasing aggregation of ESVs in the GBA. Specifically, from 2000 to 2010, the "high-high" area continued to expand southward, which is mainly due to the obvious transfer of woodland to the south of the GBA, the high vegetation cover of wetlands and river corridors, the better maintained landscape pattern and the more stable ecological land function structure. The "low-low" region is shifting from scattered distribution to patchy development, such as in Foshan, Zhongshan, Jiangmen and other cities, mainly because the concentrated development of construction land increased the intensity of land use and increased pressure on the ecological environment within the city. The Hong Kong and Macao special administrative regions also belong to the "low-low" region, mainly because of the limited space for urban development in the region, so a large number of land reclamation projects were carried out, encroaching on natural ecological land such as coastal shorelines and water areas, and the increase in the proportion of human-made land surface reduced the ESVs. From 2010 to 2020, the "high-high" zone continued to develop to the south, and the "low-low" zone is dominated by areas of concentrated urban agglomerations, which generate low ESVs due to a low intra-urban landscape diversity (Figure 9).

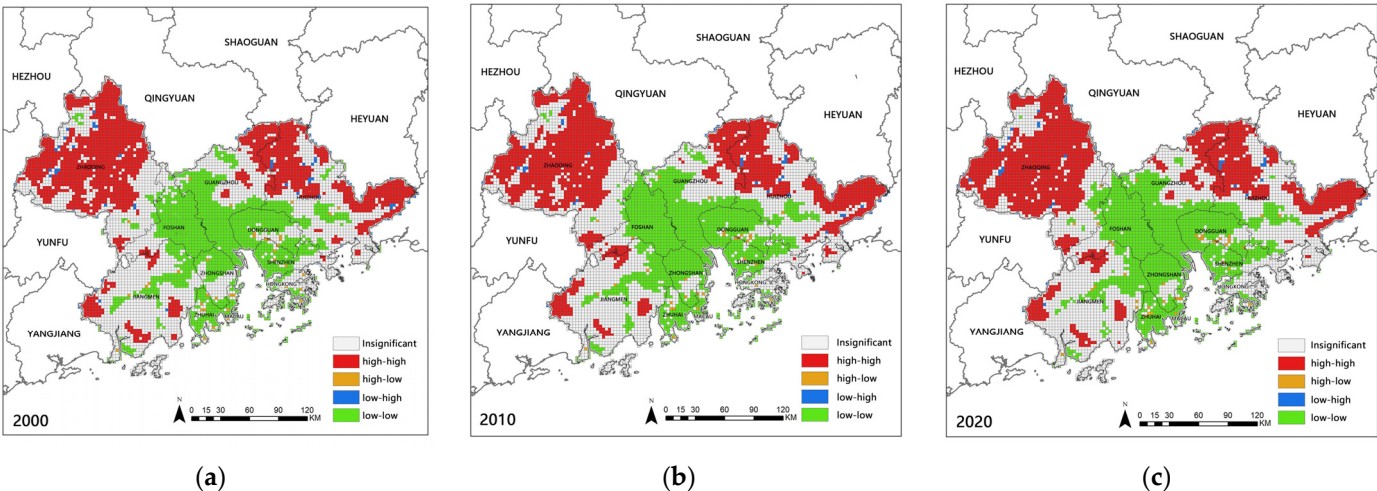

**Figure 9.** Local spatial autocorrelation of ecosystem service values in the Guangdong–Hong Kong–Macao Greater Bay Area in 2000 (**a**), 2010 (**b**) and 2020 (**c**).

### 3.2.3. Forecasting the Ecosystem Service Values

By running the Markov model under the natural development scenario, the predicted ESV in 2030 reaches $1853.921 \times 10^8$ CNY, a decrease of $64.839 \times 10^8$ CNY compared to 2020, which is mainly due to the decreasing areas of watershed, woodland and grassland from 2000 to 2020, while the area of arable land shows stability in the context of policy protection. According to this pattern of development, the areas of watershed, woodland and grassland will keep decreasing in the future, resulting in a decreasing ESV in the forecast year 2030. At the same time, it can be observed that the future decline of water resources will remain a serious problem, and therefore, scientific and rational planning are needed to reduce damage to water resources and the near-coastal environment (Table 7).

**Table 7.** Projections of ecosystem service values ($1 \times 10^8$ CNY) in the Guangdong–Hong Kong–Macao Greater Bay Area in 2030.

| Ecosystem Service Types | Ecosystem Services | Arable Land | Woodland | Grassland | Water | Unused Land | Construction Land | Total |
|---|---|---|---|---|---|---|---|---|
| Supply Services | Food production | 23.930 | 18.121 | 0.861 | 1.697 | 0.006 | 0.000 | 44.616 |
| | Raw material production | 9.333 | 163.641 | 0.721 | 1.121 | 0.012 | 0.000 | 174.828 |
| Reconciliation Services | Atmospheric regulation | 17.230 | 237.224 | 3.005 | 1.633 | 0.018 | 0.000 | 259.110 |
| | Climate regulation | 23.212 | 223.496 | 3.125 | 6.598 | 0.039 | 0.000 | 256.470 |
| | Hydrological regulation | 18.426 | 224.594 | 3.065 | 60.115 | 0.021 | −60.090 | 246.131 |
| Support Services | Waste disposal | 33.263 | 94.450 | 2.644 | 47.560 | 0.078 | −19.563 | 158.433 |
| | Soil maintenance | 35.177 | 220.750 | 4.487 | 1.313 | 0.051 | 31.314 | 293.093 |
| | Biodiversity maintenance | 24.409 | 247.658 | 3.746 | 10.985 | 0.121 | 0.000 | 286.918 |
| Cultural Services | Recreation | 4.068 | 114.219 | 1.743 | 14.220 | 0.072 | 0.000 | 134.322 |
| | Total | 189.048 | 1544.154 | 23.395 | 145.244 | 0.419 | −48.339 | 1853.921 |

### 3.3. Analysis of the Topographic Factor-Driven Effects of Ecosystem Service Value

The elevation and slope were selected as the topographic characterization factors of the study area with reference to the relevant literature. The results of the geographically weighted regression model (GWR) show that $R^2 > 0.8$ and local $R^2 > 0.8$, indicating that the model operation results meet the accuracy requirements. The results show that the local $R^2$ of the data from 2000 to 2020 changed significantly, showing a decreasing trend and then an increasing trend, indicating that the influence of topographic factors on ESVs generally show an increasing trend. The regression parameters of the topographic factors (Table 8) show that the local $R^2$ values are above 0.85 in all years, indicating a significant correlation between the topographic factors and the ESVs.

**Table 8.** Topographic Factor Regression Parameter Table for Guangdong, Hong Kong and Macau Greater Bay Area.

| Year | 2000 | 2010 | 2020 |
|---|---|---|---|
| Local $R^2$ | 0–0.8541 | 0–0.8532 | 0–0.8584 |
| $R^2$ | 0.8022 | 0.8267 | 0.8317 |
| Adjusted $R^2$ | 0.7883 | 0.8146 | 0.8200 |
| AICC | 256,353.8 | 256,973.8 | 256,797.1 |

In terms of spatial distribution, the strongest influence of topography on ESV is found in the northwestern and northeastern parts of the GBA, where the correlation with the land use data shows that woodland, arable land and water are the main land types, while the influence of the land–water interface in the south is also evident. Elevation and slope have a direct influence on plant growth, which determines the vegetation cover of the area and, thus, has a further influence on the ESVs provided (Figure 10).

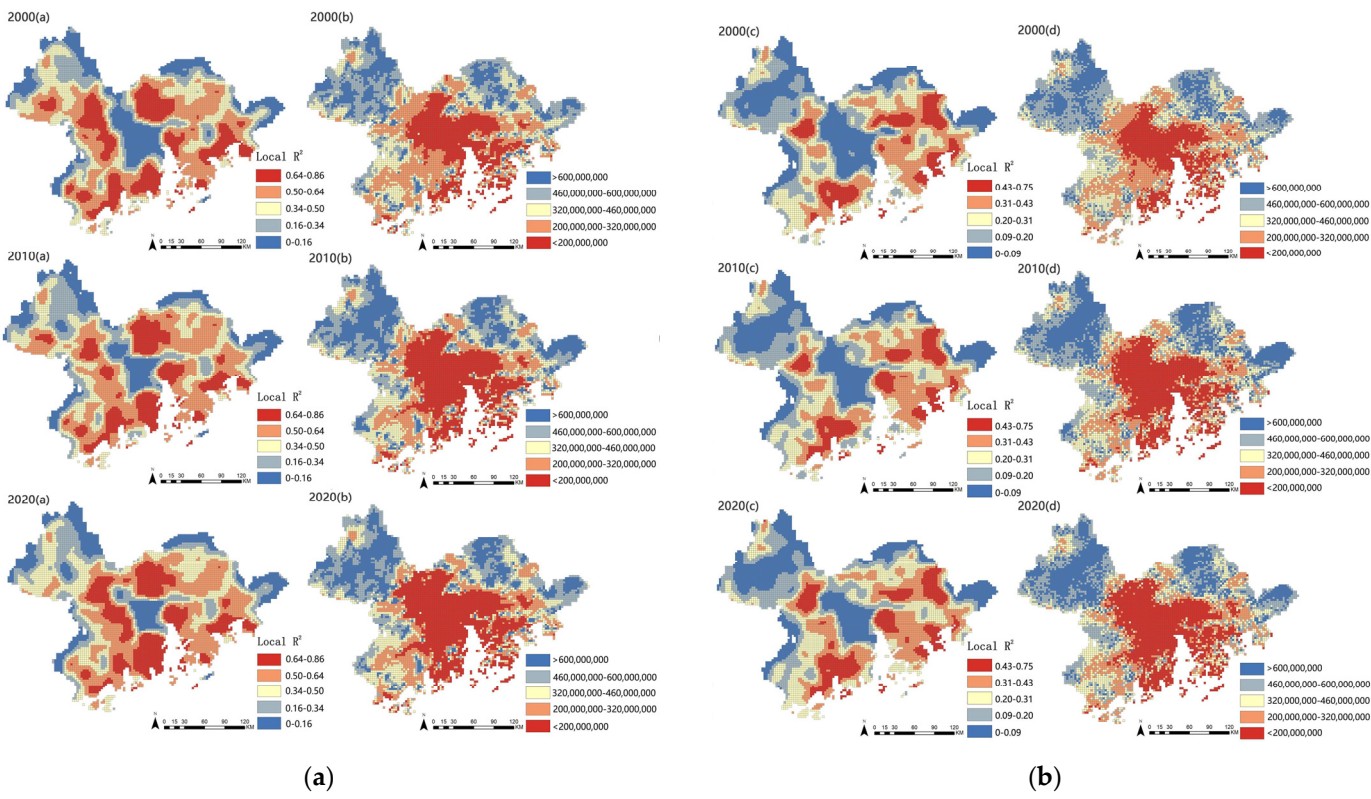

**Figure 10.** Elevation local R2 and predicted map from 2000 to 2020 (**a**) and slope local $R^2$ and predicted map from 2000 to 2020 (**b**).

From the perspective of construction land, apart from parks and green belts within the city, most areas have poor green infrastructure, and grey infrastructure, human housing and other construction land directly change the ecological landscape pattern of the land, which shows that human factors have a very strong influence on the ESVs. From 2000 to 2020, the trend of the overall pattern of the GBA clusters into patches, which shows that the influence of topographic factors on ESVs is gradually increasing.

## 4. Discussion and Conclusions

### 4.1. Discussion

4.1.1. Land Use Change Significantly Affects Ecosystem Service Values

Land is fundamental; the state and structure of the ecosystem are dependent on the land use approach, and the excessive repurposing of ecological land areas will cause the

center of gravity of the land to continuously migrate, changing the landscape pattern of the region to a large extent, while affecting the supply of ESs. To achieve the sustainable development of the regional ecosystem, we must first pay attention to the adjustment of the land use structure; the regulation capacity and carrying capacity of the natural environment need reasonable planning of land use [54]. For example, in the areas around construction sites, the construction of nature reserves, the ecological restoration of rivers and lakes and the management of land reclamation projects can be strengthened to safeguard biodiversity, soil conservation and other ecological functions so as to safeguard the carrying capacity and self-regulating ability of the natural environment as a whole and enhance ESVs [55,56].

### 4.1.2. Spatial and Temporal Variation in Ecosystem Service Values and Strategies for Optimization

The spatial and temporal variations in ESVs in the GBA were significant from 2000 to 2020. Overall, in terms of the spatial distribution, the high-value areas are concentrated in the northwest, northeast and southwest regions where ecological land is widely distributed, while the low-value areas are mainly concentrated in the urban contiguous development areas and some coastal areas in the central part of the GBA, and the area of the low-value areas is still increasing. Specifically, in order to slow down the rate of diminishing ESVs, we should take the initiative to find entry points and explore optimization strategies in terms of human social life [57–59]. In terms of water resources, while the deep treatment and re-circulation of domestic fecal water and industrial wastewater should be carried out to achieve less or no discharge and improve the utilization rate of water resources, the rational use of water resources should be strengthened, water supply and drainage systems should be improved, land use planning should be carried out scientifically and reasonably and damage to water resources and the near-shore environment should be reduced [60,61]. In terms of energy use, the characteristics of each city in the GBA should be addressed, the upgrading of equipment for renewable and clean energy and the development of new energy sources should be enhanced to provide a reasonable and effective supplement [62]. In terms of solid waste treatment, while improving the harmless treatment of waste, the whole process of management and control should be implemented from all aspects of waste generation, collection, transportation, reuse and treatment to avoid causing adverse effects on the ecological environment and achieve control of waste generation.

### 4.1.3. Topographic Factors and Ecosystem Service Values

Topography has a direct impact on the growth state and diversity of plants, while the type, growth and distribution of plants have a critical impact on key regional ESs, such as soil and water conservation and water harvesting functions. The above analysis shows that there is a significant correlation between topographic factors and ESVs in the GBA, with an increase from 2000 to 2020. In the northwestern and northeastern parts of the region, the influence of topography on ES is particularly significant in areas such as woodland, arable land and the land–water interface, so that the protection and optimization of these areas is conducive to the sustainable development of the region, to the optimization of ESVs and to the construction of ecological security patterns across the region [63].

### 4.1.4. Sustainability in the Guangdong–Hong Kong–Macao Greater Bay Area

The results of the valuation of ESs in the GBA show that the region is currently facing significant ecological degradation and intensified resource and environmental constraints. At the same time, due to the GBA being located in the subtropical monsoon climate zone, its special geographical location and climatic conditions expose it to ecological risks such as flooding, storm surges and sea level rise, which pose challenges to regional ecological restoration and the construction of ecological security patterns. In the face of the continuous decline of ecological and environmental quality in the GBA, it is necessary to gradually promote the formation of green and low-carbon production and lifestyle and urban construction and operation mode to realize the sustainable development of the GBA. Therefore,

the future development planning of the GBA should pay attention to the delineation of ecological protection zones and the control of urban expansion boundaries, and resist ecological risks such as storm surges and sea level rise by building a resilient multi-objective, cross-habitat and land–sea integrated regional ecological security pattern, while promoting the ecological restoration of coastal wetlands and coastal zone maintenance to construct a coastal security barrier in the GBA to achieve sustainable development [41].

### 4.2. Conclusions

Based on the land use changes in the GBA from 2000 to 2020, this study measures the ESVs in the GBA at the grid scale, predicts the ESVs in 2030 and analyzes its spatial and temporal evolution characteristics and the influence of topographic factors in ESVs. This will provide scientific guidance for optimizing ESVs.

(1) Over the past 20 years, in terms of the shift in the land area, the most obvious fluctuations are in the areas of water, arable land and construction land; the northwestern and northeastern areas with high vegetation cover gradually decreased, while the impervious area of the urban agglomerations in the central area is increasing, and the ecological land around the urban agglomerations, such as arable land and wetlands, is decreasing. In terms of the shift in the center of gravity of the land, the center of gravity of all types of land has shifted to different degrees, with the center of gravity of construction land shifting northward, while the center of gravity of grassland and unused land shifted westward, and the center of gravity of forest land has shifted northwestward, with significant contradictions between the survival of ecological land and the expansion of urban construction land.

(2) From an overall perspective, the ESVs in the GBA from highest to lowest are the following: regulating services > supporting services > provisioning services > cultural services. Hydrological regulation, biodiversity and soil conservation are the three items that contribute the most to the values of individual ESs. In terms of temporal changes, the ESs in the GBA showed a decreasing trend in the overall and individual values over the 20-year period. In terms of the spatial changes in rank, the area with the lowest ESV rank is located in the dense urban area in the central part of the GBA, accounting for 35% of the total area and increasing in size, indicating the deteriorating quality of the habitat; the area with a relatively high ESV rank is located in the city of Zhaoqing, northeastern Huizhou and Jiangmen in the GBA, accounting for 27% of the total area. The Markov model predicts that the ecosystem service value in 2030 shows a decreasing trend under the development of a natural state.

(3) The spatial distribution of ESVs in the GBA is aggregated, and there is a regional adjacency between the high-value and low-value areas of ESVs. From 2010 to 2020, the "high" area continues to develop to the south, and the scope of the low-value aggregation area is also expanding. The construction land in the low-value area is increasing, and the population and traffic pressure are becoming bigger and bigger, which has a great impact on the urban green space system and reduces the anti-disturbance ability of the inner-city green space ecosystem.

(4) There is a significant correlation between topographic factors and ESV, where topographic factors have a strong influence on ESVs mainly in the northwestern and northeastern parts of the GBA, and the coupling analysis with the land use data show that the main land types in this area are forest land, arable land and water. At the same time, the influence of the water–land intersection zone in the south is also obvious; the weaker influence is mainly located in the central part of the study area and the border area, and the main land types are construction land, dike-ponds and forest land.

**Author Contributions:** Conceptualization, H.L., Y.H. and Y.Z.; methodology, H.L., Y.H. and Y.Z.; software, Y.H. and Y.Z.; validation, Y.H. and S.W.; formal analysis, H.L., Y.H. and S.W.; investigation, W.G. and Y.L.; resources, J.W.; data curation, Y.H., Y.Z., J.W. and Y.L.; writing—original draft preparation, H.L. and Y.Z.; writing—review and editing, H.L., Y.H., S.W., Y.Z. and Q.X.; visualization, Y.H.,

Q.X., X.Z., K.Y., Q.H. and L.L.; supervision, H.L., Y.H. and W.L.; project administration, H.L.; funding acquisition, H.L. All authors have read and agreed to the published version of the manuscript.

**Funding:** This research was funded by a National Natural Science Foundation Project, grant number 52078222, and the Guangdong Provincial Education Department's Key Scientific Research Projects of the year 2020, grant number 2020ZDZX1033.

**Data Availability Statement:** Not applicable.

**Conflicts of Interest:** The authors declare no conflict of interest.

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
