# Peer review of "Spatial and Temporal Evolution of Ecosystem Service Values and Topography-Driven Effects Based on Land Use Change: A Case Study of the Guangdong–Hong Kong–Macao Greater Bay Area"

_sustainability, doi:10.3390/su15129691_

Round 1

Reviewer 1 Report

The paper presnts Spatial and Temporal Evolution of Ecosystem Service values and Topography-Driven Effects  Based on Land Use Change: A Case Study of the Guangdong-Hong Kong-Macao Greater Bay Area 

In the paper the authors use spatial data from ASTER GDEM.  The methodology of the work is well described and illustrated in readable figures. Research famework is clear and transparent. The statistics used in the manuscript show significant correlations. The paper characterized ecosystem services value (1x108CNY) per unit area in the Guangdong-Hong Kong-Macao Greater Bay Area In results the authors used transfer matrix of the study area for characteristics for temporal land use change characteristics The results wewre also presented in tables very clearly. The conclusions of the paper show that there is a significant correlation between topography and ESV, with the stronger influence of topography on ESV.  Moreover the main land use types are forested, arable and water areas, as shown by the coupled analysis with land use data.    Literature chosesn for the literature review is well selected and covers the themes of ecosystem services and  local problems described in the works Guangdong-Hong Kong-Macao Greater Bay Area.  

Author Response

We greatly appreciate your professional review work on our article and we will further improve the structure of this article.

Reviewer 2 Report

sustainability-2389691

Review comments

The ecosystem service values is studied in this research for an area of profound natural and marine resources, GBA. The systematic evolution patterns of the dynamic of land use and driving mechanisms of the ecosystem service value of the GBA is especially valuable information for the governance and conservation of its ecosystems. A series of scientifically sound valuation method is used to study, such as grid of landscape changes.

Several important results can effectively illustrate the spatial and temporal dynamics of land use change and ecosystem service evolution. The area of arable land and water increased and greatly fluctuated. It mostly became construction land. There is significantly deteriorating habitat quality. The study also find that the topographic factors a significant influence on the ecosystem services value in terms of spatial distribution.

            Several suggestions and comments are made here for authors’ consideration as they revise this draft. The authors are suggested to note the clarity of this paper.

1. The interpretation of the spatial change of the land use and the ecosystem services seems needs to be improved. The description and background characteristics of the different areas, such as northwest, northeast, southwest and so on, needs to be clearly defined in the section of introduction, since it is relevant background of the main findings, demonstrated in the abstract, the conclusions and other parts of this paper.

2. The study had applied sound methodology, and however, more clarified findings are expected to be provided in the sections such as abstracts and other parts of this paper.

3. The writing of this article can still be improved to become neatly expressed.

2023 05 18 sustainability-2389691

Review comments

The ecosystem service values is studied in this research for an area of profound natural and marine resources, GBA. The systematic evolution patterns of the dynamic of land use and driving mechanisms of the ecosystem service value of the GBA is especially valuable information for the governance and conservation of its ecosystems. A series of scientifically sound valuation method is used to study, such as grid of landscape changes.

Several important results can effectively illustrate the spatial and temporal dynamics of land use change and ecosystem service evolution. The area of arable land and water increased and greatly fluctuated. It mostly became construction land. There is significantly deteriorating habitat quality. The study also find that the topographic factors a significant influence on the ecosystem services value in terms of spatial distribution.

         Several suggestions and comments are made here for authors’ consideration as they revise this draft. The authors are suggested to note the clarity of this paper.

1. The interpretation of the spatial change of the land use and the ecosystem services seems needs to be improved. The description and background characteristics of the different areas, such as northwest, northeast, southwest and so on, needs to be clearly defined in the section of introduction, since it is relevant background of the main findings, demonstrated in the abstract, the conclusions and other parts of this paper.

2. The study had applied sound methodology, and however, more clarified findings are expected to be provided in the sections such as abstracts and other parts of this paper.

3. The writing of this article can still be improved to become neatly expressed.

Author Response

Thank you very much for your comments. Please refer to the attachment below for specific changes. Thank you

Reviewer 3 Report

(1)    Studying the spatial and temporal variation of regional ESV involve future-year scenarios and considering the effects of topographic factors made this research more comprehensive.

(2)    Using ESV coefficients method to evaluate ecosystem services is relatively outdated, but in this research, ESV at the grid scale was corrected by the coefficient and by the vegetation cover, this approach is acceptable.

(3)    The 3km x 3km grid was finally adopted as the base measurement unit, why, and what is the selection standard?

(4)    When GBA first appears in the main text of the article, add the full name (line 38).

(5)    “The GBA is located in the south-central part of Guangdong Province (line 130)”. Is this accurate? GBA includes Hong Kong and Macau.

(6)    Compared to the parts of Guangdong, Hong Kong and Macau have a small area, making it difficult to understand the situation in Hong Kong and Macau in the entire GBA study. Can you provide appropriate explanations and comparisons for Guangdong, Hong Kong, and Macau? Briefly.

(7)    About the result and discussion on “Topographic factors and ecosystem services value”, because this research used the ESV coefficients method, so the logic is that Topographic factors determines land use, thereby affecting ESV. In fact, in some other ESV evaluation methods, Topographic factors are often important parameters.

Author Response

(The authors gave the same response as above.)

Round 2

Reviewer 2 Report

The paper is well revised. It can be acceped now.